# Specific Binding of the α-Component of the Lantibiotic Lichenicidin to the Peptidoglycan Precursor Lipid II Predetermines Its Antimicrobial Activity

**DOI:** 10.3390/ijms24021332

**Published:** 2023-01-10

**Authors:** Irina S. Panina, Sergey V. Balandin, Andrey V. Tsarev, Anton O. Chugunov, Andrey A. Tagaev, Ekaterina I. Finkina, Daria V. Antoshina, Elvira V. Sheremeteva, Alexander S. Paramonov, Jasmin Rickmeyer, Gabriele Bierbaum, Roman G. Efremov, Zakhar O. Shenkarev, Tatiana V. Ovchinnikova

**Affiliations:** 1M.M. Shemyakin and Yu.A. Ovchinnikov Institute of Bioorganic Chemistry, Russian Academy of Sciences, 117997 Moscow, Russia; 2Moscow Institute of Physics and Technology, 141700 Dolgoprudny, Russia; 3Institute of Medical Microbiology, Immunology and Parasitology, Medical Faculty, University of Bonn, 53117 Bonn, Germany; 4Department of Applied Mathematics, National Research University Higher School of Economics, 101000 Moscow, Russia; 5Department of Bioorganic Chemistry, Faculty of Biology, Lomonosov Moscow State University, 119234 Moscow, Russia

**Keywords:** antimicrobial peptides, bacteriocins, *Bacillus licheniformis*, lantibiotics, lipid II, molecular dynamics, NMR spectroscopy, peptidoglycan, pyrophosphate cage

## Abstract

To date, a number of lantibiotics have been shown to use lipid II—a highly conserved peptidoglycan precursor in the cytoplasmic membrane of bacteria—as their molecular target. The α-component (Lchα) of the two-component lantibiotic lichenicidin, previously isolated from the *Bacillus licheniformis* VK21 strain, seems to contain two putative lipid II binding sites in its *N*-terminal and *C*-terminal domains. Using NMR spectroscopy in DPC micelles, we obtained convincing evidence that the *C*-terminal mersacidin-like site is involved in the interaction with lipid II. These data were confirmed by the MD simulations. The contact area of lipid II includes pyrophosphate and disaccharide residues along with the first isoprene units of bactoprenol. MD also showed the potential for the formation of a stable *N*-terminal nisin-like complex; however, the conditions necessary for its implementation in vitro remain unknown. Overall, our results clarify the picture of two component lantibiotics mechanism of antimicrobial action.

## 1. Introduction

Due to the increasing threat of infections caused by antibiotic-resistant pathogens, natural antimicrobial peptides have attracted the attention of researchers in recent years. Lantibiotics are among the most numerous and promising classes of antimicrobial peptides of bacterial origin. These molecules are synthesized ribosomally and then undergo significant post-translational modifications, including dehydration of the Ser and Thr amino acid residues and their subsequent cyclization with Cys residues forming lanthionine (Lan, D-Ala-Cys) and 3-methyllanthionine (MeLan, Abu-Cys) residues (Figure 1E). The antibacterial activity spectrum extends to both Gram-positive and Gram-negative bacteria, including clinically relevant pathogens such as *Listeria monocytogenes*, *Salmonella enterica*, methicillin-resistant *Staphylococcus aureus* (MRSA), and vancomycin-resistant enterococci (VRE), yet maximum efficacy is shown when they are used in combination with antimicrobial compounds of other classes [1,2,3,4]. The activity against Gram-negative bacteria is less pronounced due to low permeability of the outer membrane for lantibiotic molecules that are too large to utilize the porin pathway of translocation [5]. In-depth study of the mechanisms of action of these peptides is required for the development of effective drugs to combat various bacterial infections, including antibiotic-resistant ones.

Today, it is known that some lantibiotics are able to bind to lipid II (undecaprenyl-pyrophosphoryl-MurNAc-(pentapeptide)-GlcNAc) (Figure 1F), which is a part of the bacterial cytoplasmic membrane and the precursor in the synthesis of peptidoglycan of Gram-positive and Gram-negative bacteria. Lipid II is a target for many antimicrobial agents: non-ribosomally synthesized antibiotics (vancomycin, ramoplanin, mannopeptimycins, teixobactin), eukaryotic antimicrobial peptides (animal and fungal defensins), and also post-translationally unmodified (lactococcin 972) and modified bacteriocins [6,7,8,9,10]. Lantibiotics including nisin [11], NAI-107 (microbisporicin) [12], gallidermin [13], nukacin ISK-1 [14], mersacidin [15], haloduracin [16], and lacticin 3147 [17] have been shown to form a complex with lipid II.

**Figure 1 ijms-24-01332-f001:**
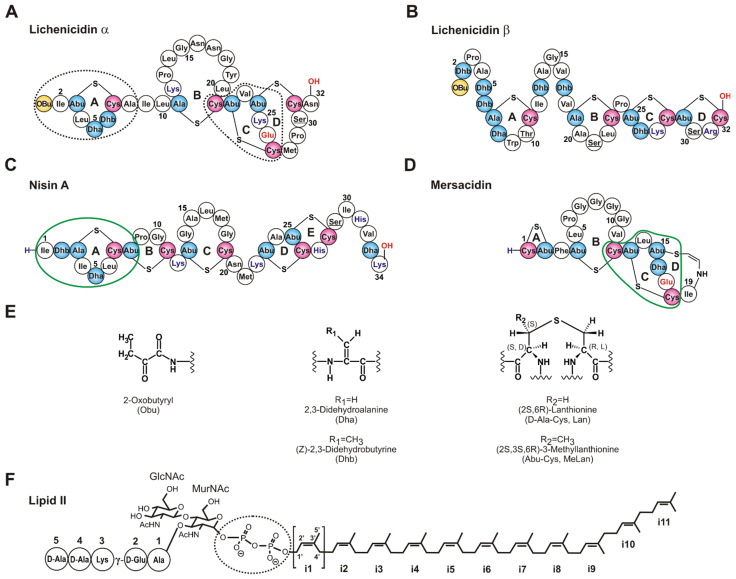
Structures of some of the lantibiotics mentioned in this work and lipid II. (**A**,**B**) The α- and β-components of lichenicidin (Lchα and Lchβ) [18]. It is worth noting that the arrangement of the ring A in Lchα differs from that in lichenicidin I89 [19]. The residues that are derived from Ser/Thr are shown in blue (Dha/Dhb/Ala/Abu), red (Cys), and yellow (Obu). Unmodified Ser/Thr are underlined. (**C**,**D**) Nisin A and mersacidin—classic examples of type A and type B lantibiotics containing different lipid II binding sites. (**E**) The chemical structures of post-translationally modified amino acid residues. (**F**) The structure of lipid II with Lys residue in the 3rd position of the pentapeptide. Isoprene repeats of the bactoprenol moiety are sequentially labeled *i1–i11*. The pyrophosphate group in lipid II, the lipid II-binding sites in nisin and mersacidin, and the putative lipid II-binding sites in Lchα are encircled.

For lipid II-binding lantibiotics, two types of antibacterial mechanisms are known. The antimicrobial activity of nisin and nisin-like lantibiotics (e.g., subtilin, ericin), having an elongated flexible structure (type A lantibiotics), combines pore formation and inhibition of cell wall biosynthesis [20]. In the case of nisin (Figure 1C), this is possible due to the presence of two different structural domains located at the *N*- and *C*-termini of the peptide. The two *N*-terminal lanthionine and methyllanthionine rings (A and B, respectively) form a “cage-like” envelope that binds the pyrophosphate moiety (PPi) of lipid II [11,21]. The formation of the peptide–lipid II complex enhances an ability of the *C*-terminal segment containing the rings D and E to penetrate the cell membrane.

Mersacidin (Figure 1D) and mersacidin-like peptides, such as plantaricin C, lacticin 481, and actagardine A, possess a more compact, globular structure (type B lantibiotics), and, hence, they are unable to form membrane-spanning complexes. Antimicrobial activities of these molecules solely depend on their ability to arrest the peptidoglycan biosynthesis [15]. In mersacidin, the residues that make up two *C*-terminal methyllanthionine rings C and D are responsible for the binding to lipid II. Among them, the conserved Glu17 is particularly noteworthy—its replacement with Ala abolishes or markedly reduces the antibacterial activity, although its role in lipid II binding remains unclear [22]. Similarly to nisin, the binding site for mersacidin-like peptides is centered at the pyrophosphate moiety of lipid II [23]. However, in this case, the binding site is larger and extends to the disaccharide group [15] or to the pentapeptide fragment [23].

Initially, the structures of lipid II complexes with peptides having nisin- or mersacidin-like binding site were determined by solution NMR in organic solvents. The complex with nisin was determined in DMSO solution [11], while the complex with lacticin 3147 A1 was studied in CD_3_OH/CD_3_CN/H_2_O (1:1:1) mixture [23]. These isotropic low polarity media weakly model the membrane environment, which is essentially anisotropic and has hydrophobic and polar regions. As a result, the obtained structures raised some criticism. For example, a subsequent solid-state NMR study has conclusively shown that the conformation of the lipid II-bound nisin was drastically different in membranes [24]. Moreover, a recent computational study provided an alternative model of nisin/lipid II complex in the membrane [21]. Presently, there are no structural data on complexes of mersacidin-like peptides with lipid II in the membrane.

In addition to the above mentioned single-peptide lantibiotics, there is a group of two-component systems, currently numbering 13 representatives, each consisting of two separately produced peptides (α- and β-), which are able to exhibit much higher activity in synergy [25,26,27,28,29,30,31,32,33]. For two-component lantibiotics, both described mechanisms of antibacterial action are possible. The globular α-subunit containing the mersacidin-like motif is responsible for binding to lipid II, while the elongated β-subunit interacts with this complex in an equimolar ratio and forms pores in the cell membrane [34,35,36].

Among the currently known natural lantibiotics, there are no peptides with the confirmed presence of two lipid II binding sites in one molecule. In this respect, the α-component of lichenicidin (Lchα), as part of a synergetic lantibiotic pair isolated from the *Bacillus licheniformis* VK21 strain (Figure 1A,B), is of particular interest [18]. In addition to the mersacidin-like motif in the *C*-terminal part, which is conserved among the α-peptides of two-component lantibiotics, it has the *N*-terminal nisin-like motif (Figure 1A). Superimposition of the Lchα spatial structure with that of the nisin-lipid II complex revealed at least three hydrogen bonds (compared to five bonds observed for nisin [11]) that could be formed between backbone amides of Lchα and oxygen atoms of PPi, although several sterical clashes were also noted [18]. At the same time, comparison of the Lchα structure with the structure of mersacidin showed a close similarity of the *C*-terminal domains including the spatial position of the charged Glu residue and the presence of a flexible hinge between the rings B and C.

Recent carboxyfluorescein leakage experiments with liposomes of different composition and lichenicidin I89 (a natural variant of lichenicidin isolated from the *Bacillus licheniformis* I89 strain and composed of Licα and Licβ) have pointed to the role of lipid II as a landing site for Licα, as was previously proposed for other lantibiotics [37]. The aim of this work was to elucidate the mechanism of lipid II binding by Lchα and to reveal roles of both putative binding sites using in vitro and in silico approaches. We studied Lchα/lipid II interactions in the anisotropic membrane mimicking environment provided by detergent micelles and modeled the complexes in the membrane. Here, we demonstrated that the residues of the rings C and D are involved in lipid II binding, and Ca^2+^ ions may play some role in the membrane activity of Lchα. The *N*-terminal binding site under some conditions might also participate in the interaction with lipid II.

## 2. Results

### 2.1. The Impact of Divalent Cations on the Antibacterial Activity of Lchα

Antimicrobial activities of lichenicidin and its α-component against two strains of Gram-positive bacteria were investigated in half Mueller–Hinton broth (½ MH) without salts and with the addition of Ca^2+^ and Mg^2+^ ions. The growth rate of *B. megaterium* in the presence of the individual Lchα showed a marked decrease after the addition of 1.25 mM Ca^2+^ (Figure 2). At the same time, Ca^2+^ did not affect the level of activity of the equimolar mixture Lchα:Lchβ. When the cultures were grown in the presence of Mg^2+^ ions, the cell density was higher than in the medium without additives, including the control experiments without antimicrobial peptides. Probably, Mg^2+^ plays the role of a growth factor that is deficient in the standard ½ MH. For comparison, we used the β-hairpin peptide AA139, a typical salt-sensitive cationic antimicrobial peptide with a low-specific membranolytic mode of action [38]. Experiments with AA139 showed that both divalent cations protect *B. megaterium* from the peptide action, with Ca^2+^ being more efficient than Mg^2+^. In the case of *M. luteus*, Ca^2+^ did not affect antibacterial activities of the tested peptides, while Mg^2+^ addition led to a slight growth enhancement.

### 2.2. NMR Study of Lchα in Water and Methanol

In our previous study, we have shown that Lchα is soluble in isotropic solvents ranging from polar H_2_O/CD_3_OH (3:1) to strongly hydrophobic CD_3_OH/CDCl_3_ (2:3) [18]. Here, we determined that the peptide is well soluble in water and studied ^15^N-labeled Lchα in this environment (Figure 3A). The HN-groups of all non-proline residues of the peptide were observed in the ^15^N-HSQC spectrum, but a significant heterogeneity was present. Most of the signals had at least three components shown in black, red, and blue. This is consistent with the data obtained previously in methanol solution [18]. The observed heterogeneity is probably a consequence of conformational exchange processes going slow on the NMR time scale (characteristic time >> 50 ms). These processes can be cis–trans isomerization of the Lys12–Pro13 and/or Met28–Pro29 peptide bonds or fluctuations in the conformation of the thioether rings. The quality of the obtained spectra does not allow us to distinguish between these possibilities; however, the strong H^α^_i_–H^δ^_i+1_ NOE cross-peaks observed for Lys12–Pro13 and Met28–Pro29 dipeptide fragments indicated that corresponding peptide bonds have trans configuration in the major Lchα conformation.

The almost complete ^1^H,^13^C,^15^N backbone and side chain resonance assignment was obtained for the major Lchα conformer in water at pH 4.0 and 30 °C (Appendix A). In addition, the complete ^15^N resonance assignment was obtained for the major Lchα conformer in methanol (pH 3.5, 27 °C; Appendix A).

The Lchα peptide contains four (methyl)lanthionine rings that belong to three domains: the *N*-terminal domain (Obu1–Leu10, the ring A), the central domain (D-Ala11–Cys21, the ring B), and the *C*-terminal domain (Abu22–Asn32, the rings C and D) (Figure 1A). Lanthionine and methyllanthionine residues contain two and three chiral centers, respectively (Figure 1E). According to the published data, the most common configuration of Lan residues in lantibiotics have the (2*S*,6*R*) or DL configuration, while (2*R*,6*R*) or LL stereochemistry was described in the specific cases [39]. Similarly, the majority of MeLan residues have (2*S*,3*S*,6*R*) configuration, and the (2*R*,3*R*,6*R*) and (2*S*,3*R*,6*R*) stereochemistries were described in specific cases [40]. In our previous study [18], we, by chance, have used the (2*S*,3*R*,6*R*) configuration for MeLan residues in Lchα, and the erroneous chirality around the Cβ atom of Abu residues (*R* instead of *S*) was adopted. Therefore, here, we recalculated previously obtained spatial structure of Lchα in methanol. The set of 20 Lchα structures was calculated in the CYANA program using published structural restraints (NOE-based distance restraints, torsion angle restraints, and thioether bonds restraints). The representative structure is shown in Figure 4A, while a set of 20 Lchα structures superimposed on individual domains are presented in Appendix A. The results of structure calculations (Appendix A) revealed that the experimental NMR restraints are consistent with the correct geometry of Abu-containing methyllanthionine rings. At the same time, the change in Abu C^β^ atom configuration led to significant changes in the calculated structures, especially in the *C*-terminal domain (Appendix A). Backbone atoms’ root-mean-square deviation (RMSD) (Appendix A) shows that the obtained structure matches well along the *N*- and *C*-terminal domains. However, the central domain has no fixed conformation (Appendix A). The domains are probably connected by flexible hinges at residues D-Ala11 and Cys21.

Comparison of ^1^H^15^N and ^1^H^α^ chemical shifts of Lchα in water and methanol (Figure 4C,D) revealed significant differences (Δδ^1^H^15^N > 0.5 ppm) only for residues Ile9–Leu10, located between the rings A and B. Probably, the structure of Lchα does not change much upon the peptide transfer from water to a more hydrophobic methanol environment, and the changes are limited to the *N*-terminal domain and its connection with the central domain. The large (Δδ^1^H^α^~0.4 ppm) change of chemical shift of the *C*-terminal Asn32 residue (Figure 4D) can be explained by the difference in the ionization state of the *C*-terminal carboxyl group in methanol (pH 3.5) and water (pH 4.0).

### 2.3. NMR Study of Lchα in DPC Micelles

It is assumed that the target for Lchα is lipid II [18]; thus, the active conformation of the peptide can be observed only in the lipid membrane or some membrane-mimicking environment. We used micelles of the zwitterionic detergent DPC to mimic the membrane environment under conditions suitable for the high-resolution NMR study. Similar to biological membranes, the micellar environment is anisotropic, i.e., demonstrates separation between hydrophobic and polar regions.

To characterize binding of Lchα to DPC micelles, the changes of the peptide chemical shifts were monitored upon gradual addition of the detergent to the Lchα sample (Figure 4B). The chemical shift of residues from the *N*- and *C*-terminal domains demonstrated similar behavior: a sharp change at the initial steps of DPC titration and a plateau after the detergent to peptide molar ratio (D:P) reached up to 30:1. This indicates a complete transition of the Lchα molecules to the micelle-bound state. Interestingly, some signals of the *C*-terminal domain (e.g., HN of Abu24, Figure 4B) were not observable at D:P ratio of ~8:1, while full titration curves were detected for the *N*-terminal residues (e.g., Cys7). This indicates that binding of *N*- and *C*-terminal domains to the DPC micelle are independent processes having different time scales. Nevertheless, both titration curves were nicely approximated by the single Langmuir equation with the following parameters: free energy of the complex formation (Δ*G*^0^ = −6.5 ± 0.4 kcal·mol^−1^, *K_N_* = (4.6 ± 3.2)·10^4^ mol^−1^) and the number of DPC molecules that form the Lchα binding site on the micelle surface (*N* = 9.7 ± 1.1). The obtained Δ*G*^0^ value is comparable to the values observed for membrane active antimicrobial peptides [41] and indicates a relatively high affinity of Lchα to the DPC micelle. At the same time, a relatively large value of the parameter *N* revealed that this high affinity interaction is achieved by summation of multiple weak interactions with individual DPC molecules. This pool of DPC molecules, probably, independently solubilizes different domains of Lchα.

The significant conformational heterogeneity was evident from the ^15^N-HSQC spectrum of Lchα in DPC micelles (Figure 3C). Similar to the situation observed in water and methanol, most of the HN signals had at least three components (Figure 3C, black, red, and blue). Interestingly, the signals of the residues near the D-Ala11–Cys21 lanthionine bridge and Pro13 (D-Ala11, Leu14, Leu20, Cys21, Abu22) were broadened and weakened. This indicated that some of the slow millisecond processes of conformational exchange in the Lchα molecule in a solution of DPC micelles have a faster time scale than in water and are closer to the intermediate regime.

The almost complete ^1^H,^13^C,^15^N backbone and side chain resonance assignment was obtained for the major Lchα form in DPC micelles at pH 5.8 and 45 °C (Appendix A). The ^13^C chemical shifts of C^β^ and C^γ^ atoms of Pro29 residue confirmed that the Met28–Pro29 peptide bond has trans configuration in the major Lchα conformer.

Comparison of ^1^H^15^N and ^1^H^α^ chemical shifts of Lchα in water and DPC micelles (Figure 4E,F) revealed more pronounced structural changes as compared to those in methanol. The large differences (Δδ^1^H^15^N > 0.5 ppm or Δδ^1^H^α^ > 0.15 ppm) were observed mainly in the two regions: the junctions between the *N*-terminal, central, and the *C*-terminal domains (Ile8–Lys12 and Leu20–Lys25). Interestingly, both regions contain positively charged Lys residues, which can interact with negatively charged phosphate groups of the detergent. (It is of note that the Lys side chain does not change its ionization state upon variation of pH from 4.0 to 5.8). These results suggest that solubilization of Lchα in anisotropic environment of DPC micelles does not change much the structure of the peptide domains, but mainly affects their dynamics, relative orientation, and interdomain interactions.

In addition to NMR, we performed MD simulations of Lchα interaction with the model bacterial membrane composed of POPG and POPE lipids at a ratio of 3:1 (Appendix A). In all simulations (three 500-ns replicas) the peptide adsorbed onto the water–lipid interface, generally confirming the membrane activity.

### 2.4. Backbone Dynamics of Lchα in Water, Methanol, and DPC Micelles

The use of the ^15^N-labeled analog of Lchα and ^15^N-relaxation measurements (Figure 5 and Appendix A) allowed us to characterize both the overall rotational diffusion of the major Lchα conformer in solution and the motions of the peptide backbone on two time scales: ps–ns and µs–ms. To characterize overall diffusion of the peptide, the apparent rotational correlation times τ_R_ were calculated for individual Lchα residues from the ratio of relaxation rates (R_2_/R_1_) (Figure 5A). In line with the expectations, the average τ_R_ value demonstrated a significant increase upon the peptide transfer from water (τ_R_ = 2.7 ± 1.1 ns, 30 °C) to DPC micelles (τ_R_ = 9.0 ± 1.6 ns, 45 °C). This corresponds to slowing of the peptide rotational diffusion due to its association with a large particle (micelle). Surprisingly, data obtained in methanol solution also revealed elevated τ_R_ values (5.5 ± 1.9 ns, 27 °C), which can be explained either by the peptide oligomerization or by significant µs-ms exchange contributions (R_EX_) to the R_2_ rates. Indeed, analysis of R_1_ × R_2_ products (Figure 5B) revealed the values that significantly exceed the maximal allowed limit (17 s^−2^ at 800 MHz) [42]. Thus, the Lchα backbone experiences exchange motions on the µs–ms time scale, and many HN groups in methanol and DPC micelles have significant R_EX_ contributions to the R_2_ rates. These contributions have a larger amplitude in methanol, which indicates that in this medium, the corresponding slow (on the NMR time scale) exchange process has a faster time scale than in water and DPC micelles and is closer to the intermediate regime. Most probably, it is these motions that are responsible for the conformational heterogeneity of Lchα observed in various media.

The values of the nuclear Overhauser effect between the ^1^H and ^15^N nuclei (^15^N-{^1^H}-NOE, Figure 5C) allowed us to qualitatively characterize the ps–ns time scale motions of the Lchα backbone. In water, an increased mobility on this time scale (NOE < 0.6) was observed in all three domains of the peptide. The largest mobility was observed in the *N*-terminal domain and for the *C*-terminal Asn32 residue (NOE values from −0.2 to 0.2), while the residues at the domain boundaries (Leu10–Lys12 and Abu22–Glu26) were most stable (NOE ~ 0.5–0.6). The transfer of Lchα to methanol or DPC micelles results in stabilization of the peptide backbone on the ps–ns time scale. The most pronounced stabilization was observed for the residues of the *N*-terminal domain. The peptide in the DPC micelles showed the highest “stability”. However, even in this environment, the mean NOE value (0.59 ± 0.16) indicated significant ps–ns motions. A more quantitative analysis of the ^15^N relaxation data using, for example, a “model-free” approach was impossible due to the presence of extensive conformational exchange.

### 2.5. NMR Investigation of Lchα/Lipid II Interaction in DPC Micelles

To investigate the binding of lipid II to Lchα, the peptide sample in DPC micelles was titrated with lipid II to a molar excess of 1:4 (Figure 3D). Comparison of the ^15^N-HSQC spectra of Lchα in the presence and absence of lipid II revealed significant changes in the intensity of signals of the major peptide conformation (Figure 6A). At the same time, the signals of the minor conformers remain almost unchanged. No significant changes of the peptide chemical shifts were observed upon lipid II titration (maximal Δδ^1^H^15^N—0.06 ppm). The only exception is the signal of the *C*-terminal Asn32 residue (Figure 6A). In this case, a gradual decrease in the intensity of the ^1^H^15^N signal of the major Lchα form upon the addition of lipid II was accompanied by an increase in a new ^1^H^15^N signal, possibly belonging to the lipid II-bound form of the peptide (Figure 6A,B). The observed effects are consistent with binding of lipid II to Lchα, where the process of exchange between the free and bound peptide is slow–intermediate on the NMR time scale. This process is faster than the process of exchange between different Lchα conformations. Moreover, only the major Lchα form (conformation) can interact with the lipid II. The simultaneous observation of the Asn32 HN signals in the major free and the lipid II-bound Lchα forms made it possible to estimate the characteristic time of the lipid II binding process. In the limit of slow chemical exchange, the characteristic time of the exchange process should significantly exceed the reciprocal of the difference between the ^1^H resonance frequencies of these two peaks (τ_EX_ >> 1/(0.2 ppm) = 1/(160 Hz) ≈ 6 ms, at 800 MHz, Figure 6A,B). Similar conclusion can be drawn from the difference in ^15^N frequencies of these two peaks (τ_EX_ >> 1/(2.5 ppm) = 1/(200 Hz) = 5 ms, at 80 MHz).

The equilibrium dissociation constant of Lchα/lipid II complex (*K_D_*) was determined from the dependence of the intensity of the ^1^H^15^N Asn32 signal of free Lchα on the concentration of lipid II (Figure 6B). It was assumed that at the used molar ratio of detergent to peptide (>30:1), all peptide molecules are bound to micelles. It was also assumed that detergent molecules form “continuous solvent” for the reaction of lipid II binding [43]. In this case, *K_D_* is inversely proportional to DPC concentration (see Equation (3) in Section 4.4). The obtained *K_D_* value ((1.0 ± 0.1)·10^−3^) corresponds to the free energy of the complex formation Δ*G*^0^ = −4.4 ± 0.1 kcal·mol^−1^, indicating a moderate stability of the Lchα/lipid II complex. Interestingly, under the conditions of the NMR experiment ([DPC]~10 mM), the apparent dissociation constant of the Lchα/lipid II complex (*K_app_*~10 μM, assuming that the compounds interact in the aqueous phase) provides a somewhat larger Δ*G*^0^ value (−7.3 kcal·mol^−1^).

The fitting of the experimental data shown in Figure 6B (*blue circles*) also provides concentration of the active Lchα conformation, the fraction of the peptide that can interact with lipid II. The calculated value (0.16 mM) corresponds approximately to half of the total peptide concentration (0.3 mM). This is consistent with the concentration of the major Lchα conformer, which also represents about half of the total peptide concentration. The calculated concentration of the Lchα/lipid II complex (Figure 6B, *red curve*) corresponds well to the intensity of the ^1^H^15^N signal from the lipid II-bound form of Lchα (Figure 6B, *red diamonds*) given the larger linewidth (the lower intensity) of this signal. This independently confirms the proposed interpretation of the spectral data.

The attenuation of Lchα signals induced by lipid II is uneven and has different amplitudes in different parts of the peptide (Figure 6C). The *C*-terminal domain demonstrates larger attenuation compared to the *N*-terminal and central domains. Several residues of the *C*-terminal domain (e.g., Val23, Abu24, Lys25, Cys31) were attenuated almost to zero intensity (Figure 6A). Assuming that residues close to the binding site show stronger signal attenuation than non-interacting residues, we can localize the lipid II binding site at the *C*-terminal domain of Lchα.

To determine the Lchα binding site on the lipid II molecule, we investigated unlabeled lipid II in the solution of DPC micelles. Nearly complete ^1^H,^13^C resonance assignment was obtained for the pentapeptide fragment, GlcNAc and MurNAc carbohydrate groups, and the first five and last 11th isoprene repeats of the bactoprenol fatty tail of lipid II in this environment (pH 5.8, 45 °C, Appendix A). The assignment of some of the signals is shown in the 2D TOCSY spectrum (Figure 7A). Two sets of MurNAc, GlcNAc, and Ala1 signals were observed in the spectra. This heterogeneity was due to the presence of a GlcNAc-MurNAc-pentapeptide impurity in the sample, which does not contain a bactoprenol pyrophosphate moiety. The mass spectrometry confirmed the presence of this impurity in the lipid II sample (Appendix A).

Comparison of the TOCSY spectra of lipid II in the presence and absence of Lchα revealed changes in the intensity of some signals (Figure 7A,B). At the same time, no significant changes of the chemical shifts were observed (maximal Δδ^1^H—0.02 ppm). The attenuation of the lipid II signals gradually increased from the *C*-terminus of the pentapeptide fragment to the *N*-terminus and carbohydrate groups (Figure 7B). The isoprene repeats of the bactoprenol moiety also showed a gradual increase in signal attenuation from the terminal repeat (*i11*) to the first repeat (*i1*). Thus, the greatest attenuation was observed for the MurNAc group and the first isoprene repeat, which surround the pyrophosphate fragment (Figure 7B). The signals of the impurity demonstrated very little attenuation (if any), so GlcNAc–MurNAc–pentapeptide is not able to bind the lantibiotic on its own. Comparison of the chemical shift values and signal intensities of the Lchα/lipid II complex with the addition of Ca^2+^ cations (3.5 mM) showed no significant changes in the complex conformation (data not shown).

### 2.6. Molecular Modeling of Pyrophosphate Recognition by Lchα

Given high sequence similarity of the *N*-terminal part of Lchα to nisin, one could not exclude interaction of this part of the peptide with the lipid II molecule. Previously, we computationally assessed nisin’s ability to interact with the pyrophosphate moiety of lipid II by performing MD-based free docking of the peptide with the lipid II mimetic—dimethyl pyrophosphate ion (DMPPi)—in water solution [21] and subsequently confirmed similar binding modes for the related epidermin and gallidermin peptides [44]. Here, analogously, we performed a series of MD simulations of Lchα in water with randomly placed DMPPi ion(s), which spontaneously bind to the peptide in either *N*- or *C*-terminal modules (Figure 8A), resembling nisin and mersacidin PPi binding motifs, respectively. During simulations, we calculated intermolecular hydrogen bonds as a measure of the “strength” of the interaction (Figure 8B,C and Appendix A). The structure of the *N*-terminal complex looks like nisin capturing PPi of lipid II (Appendix A), retaining DMPPi by multiple backbone hydrogen bonds—a picture that we previously observed in the nisin and epidermin/gallidermin studies [21,44]. Most of the resulting *N*-terminal complexes are very similar.

In contrast, the *C*-terminal complex structure correlates less with the presumed mersacidin motif of lipid II capture: Glu26 does not seem to interact with DMPPi (neither directly, nor via Ca^2+^-bridges), although oppositely-charged Lys25 side chain readily takes part in the interaction, provided there is an excess of DMPPi. We tried to investigate the role of Ca^2+^ in the interaction of the *C*-terminal module of Lchα with DMPPi but failed to identify any relevant events in the MD simulations (see Section 4.5), perhaps because of the force field inaccuracy for such an ion (data not shown). For this work, we performed two MD sub-series: with one and three DMPPi ions. While the *N*-terminal site operates efficiently in both cases, the *C*-terminal site unexpectedly gets involved only when the *N*-terminal site is already occupied (compare right parts of Figure 8B,C). Hence, to avoid probable MD bias in preference for charge interactions or hydrogen bonds, we do not draw a conclusion about which site is “stronger”. Although, it is worth noting that, during the described MD-based free docking, we obtained very similar *N*-terminal complexes, while there was a diversity in the *C*-terminal complexes. Figure 8A (*right*) shows only one of the probable complexes, while others are not shown.

### 2.7. Predicted Structure of Lchα/Lipid II Complexes in the Model Bacterial Membrane

Lantibiotics recognize their target—lipid II—in the membrane environment; thus, the described complexes with DMPPi in the water solution should be considered as preliminary. The stability of the full-size systems, where Lchα captures lipid II in the context of the bacterial membrane, should be assessed. Starting from the snapshots of the most populated clusters of the *N*- and *C*-terminal complexes of Lchα/DMPPi (for example, see Figure 8A), we reconstructed the complexes with lipid II. The complete lipid II molecule was rebuilt, and the complex was immersed into a model bacterial membrane in order to place the hydrophobic tail of lipid II in the membrane and leave its polar head complexed with full-length Lchα at the surface (see Materials and Methods for details). After that, a series of relaxation steps was performed, and several 500-ns MD simulations were calculated, starting from different initial conformation and/or atomic velocities (see Section 4.5).

All the studied *N*-terminal complexes were very similar in terms of their structure and pattern of intermolecular hydrogen bonds that characterize the Lchα/lipid II interaction (Appendix A). This interaction resembles much lipid II capture by nisin_1–12_ [21] and also DMPPi recognition by the three representative type A lantibiotics (nisin, epidermin, and gallidermin) [44]. More specifically, the *N*-terminal site of Lchα captures the PPi moiety by multiple hydrogen bonds, donated by the backbone amide groups of the *N*-terminal ring A, while ring B interacts with the ring A by two hydrogen bonds and stabilizes it in the recognition-competent conformation. This structure remained perfectly stable during most MD runs (see Figure 9B *left* and *middle panels* for an example), further supporting the previously proposed general mechanism for lipid II recognition by type A lantibiotics [44], which may also be shared by the *N*-terminal fragment of Lchα.

*C*-terminal complexes that remained stable during MD (Appendix A) were much more diverse and dissimilar from the *N*-terminal complexes: there are no compact group of intermolecular hydrogen bonds from the peptide backbone and a special ring structure that captures PPi, but rather a multitude of interactions that vary from complex to complex and switch during MD (Figure 9A and Appendix A). There are several common features for all observed *C*-terminal complexes in our MD studies: (1) the conserved Glu26 residue does not participate in the intermolecular contacts; (2) the main complex-forming residues are 30–32, yet, in two out of three trajectories, the Lys25 side chain forms hydrogen bond and/or salt bridge with lipid II; (3) besides PPi, many lipid II groups take part in the interaction (see Figure 9A, *middle panel*). This agrees with the data that mersacidin-like binding sites recognize not only the PPi moiety, but also other regions of lipid II, e.g., disaccharide group or pentapeptide fragment [15,23].

## 3. Discussion

In this work, we report the results of NMR study of Lchα, the α-component of the two-component lantibiotic isolated from the *B. licheniformis* VK21 strain. All obtained NMR data are consistent with the chemical structure of the mature Lchα determined previously [18] (Figure 1A). The *N*-terminal domain of Lchα contains methyllanthionine bridge formed between Abu3 and Cys7 residues (ring A) and two sequential dehydrated residues Dha5 and Dhb6. This structure is supported by sequential (H^N^_i_–H^N^_i+1_, H^α^_i_–H^N^_i+1_, H^β^_i_–H^N^_i+1_, H^γ^_i_–H^N^_i+1_) NOE cross-peaks observed for the Ile2-Abu-Leu-Dha-Dhb-Cys7 fragment, as well as by ^1^H,^13^C, and ^15^N NMR spectra, where signals of only two dehydrated residues, Dha and Dhb, were observed (data not shown).

Our structure of Lchα differs from the published structure of Licα, the α-component of the two-component lantibiotic isolated from another strain of *B. licheniformis*, I89, determined by MS-MS. Despite the identity of the amino acid sequences of the precursor and the same dehydration sites, the *N*-terminal domain of Licα contains the Dha5–Cys7 lanthionine bridge and two Dhb residues (in positions 3 and 6). Surprisingly, the β-components of the two lantibiotics (Lchβ and Licβ) are identical and exhibit synergistic effects with both α-peptide variants [19,37,45,46].

It has previously been suggested that a minor difference in the cyclase activity of LanM enzymes from the *B. licheniformis* VK21 and I89 strains (which have six amino acid mismatches) could cause the structural variability of the mature peptide [19]. Now, this assumption seems less plausible, since Lchα for this work was isolated from the strain B-511, which is the equivalent of strains ATCC14580 and DSM13. The difference in the LanM primary structure between ATCC14580/DSM13/B-511 and I89 strains is only one residue near the *N*-terminus of the enzyme (Leu21Phe), which we confirmed by the whole-genome sequencing of B-511 strain (data not shown). As we know, the arrangement of the Licα thioether bonds does not depend on whether the gene cluster is expressed in the cytoplasm of the native producer or under heterologous conditions in *E. coli* BL21(DE3) gold [19]. Thus, the heterologous environment is not the reason for the emergence of a new variant of the structure.

It is currently unknown which of the two variants of the α-peptide has a higher biological activity (independently and in a synergy with the β-peptide), but the presence of Abu3-Cys7 cyclization makes Lchα unique among other lantibiotics. This *N*-terminal structure results in formation of the nisin-like “pyrophosphate cage” for lipid II binding, that complements the *C*-terminal mersacidin-like lipid II binding site formed by the rings C and D. The NMR data about structure and dynamics of Lchα were obtained in water, methanol, and anisotropic membrane mimetic, DPC micelles. The transition of the peptide from water to a more hydrophobic methanol solution affects the conformation of the molecule to a lesser extent than its solubilization in the micelles. However, in both cases, the changes affect mainly the residues that form interdomain junctions, and not the structure of the *N*- and *C*-terminal Lchα domains. Under all conditions used for NMR study, the central domain of the peptide (D-Ala11-Cys21, ring B) probably did not have an ordered regular structure and effectively acts as a large hinge between the two putative lipid II binding sites. Most probably, such a structural organization of Lchα is also preserved in a real biological membrane. Interestingly, the presence of a hinge linker between two ordered domains is common for the type A single peptide lantibiotics. For example, Asn20-Met21-Lys22 tripeptide in nisin facilitates the insertion of the *C*-terminal domain into lipid bilayer upon interaction of the *N*-terminal domain with lipid II [20,47,48].

The backbone dynamics of the Lchα also shows marked similarity in the different environments (Figure 5). Despite the fact that the stability of the *N*-terminal and central domains in the ps-ns time scale gradually increases with the peptide transfer from water to methanol and micelles, a specific pattern of extensive exchange fluctuations on the ms time scale remains intact. It should be stressed that observed doubling (or even quadrupling) of the NMR signals cannot be ascribed to chemical inhomogeneity. The MALDI-TOF MS analysis (Appendix A) did not reveal the presence of impurities or incompletely processed peptides, while the NMR data agree with the same primary structure for major and minor Lchα components.

During the NMR study of the Lchα interaction with lipid II in the DPC micelles environment, the most significant attenuation of ^1^H-^15^N-HSQC signals was observed in the *C*-terminal domain of the peptide. This suggests that the primary lipid II binding site is in the Lchα region spanning the rings C and D (residues Abu22-Asn32, mersacidin-like site). Meanwhile, the site of Lchα binding in the lipid II molecule is centered at the pyrophosphate fragment and comprises the first few isoprene repeats from the fatty tail and the disaccharide region. Our results are consistent with the earlier suggestion of the involvement of the disaccharide headgroup in mersacidin–lipid II binding [15]. However, the interaction of the lantibiotic with the pentapeptide fragment of lipid II, described in the NMR study of the α-component of the two-component system lacticin 3147 [23], was not observed in the case of Lchα.

The results of molecular modeling of the complexes formed by Lchα with the lipid II mimetic, dimethyl pyrophosphate, and full-size lipid II partially agree with the data obtained by NMR. Variants of the *C*-terminal binding model are characterized by considerable diversity, yet reveal no interaction via Cys27, Met28, and Pro29 residues of Lchα in bilayer (Figure 9A and Appendix A), whereas residues Cys27 and Met28 were involved in lipid II binding in experimental data (Figure 6C). At the same time, the NMR and MD data are consistent in that the binding site in the lipid II molecule is not limited to pyrophosphate and includes the nearest groups. In contrast to NMR experiments, MD simulations also indicate the possibility of the formation of the “*N*-terminal complex” resembling those with nisin and nisin-like peptides (Appendix A), which is characterized by a much less structural variability.

The reason of discrepancy between NMR and MD data about the *N*-terminal binding site is presently unclear. First of all, we should mention that in our in silico study, the *N*-terminal 2-oxobutyryl group of Lchα was replaced by the conventional Val residue. Thus, the N-terminus of the peptide was supplemented by artificial positively charged -NH_3_^+^ group, which can attract negatively charged pyrophosphate moiety of lipid II. Indeed, this artificial charged group contacts the phosphate group in some of the complexes (see Figure 8A), but other complexes remained stable even without such interaction (see Figure 9B). Interestingly, nisin and nisin-like peptides contain an unblocked *N*-terminus, and the corresponding amino group interacts with the pyrophosphate moiety of lipid II (Appendix A) [21]. Second, the *N*-terminal binding site may have different properties in the real membrane (micelle) and in silico. For example, the binding site may be occupied by the phosphate group of the lipid (detergent), which prevents the binding of lipid II.

It is well known that the antibacterial and antifungal activity of many cationic antimicrobial peptides is often decreased or neutralized in salt-containing media. Cations weaken the electrostatic attraction of peptides to negatively charged components of the microbial membrane [49,50]. On the other hand, it has been shown that some mersacidin-like (but not nisin-like) lantibiotics have Ca^2+^-dependent mechanism of antimicrobial action. An average twofold increase in the mersacidin activity against several dozen strains of *Staphylococcus aureus*, *S. epidermidis*, and *Streptococcus pneumoniae* and a 7.6-fold increase against strains of *Listeria monocytogenes* have been observed in the presence of 1.25 mM Ca^2+^ [51]. Under the same conditions, antimicrobial activities of mersacidin, plantaricin C, and of the two-component lantibiotic lacticin 3147 were 2–3-fold enhanced against the *M. luteus* DSM 1790 strain [52]. Supplementation with Ca^2+^ ions also exerted positive effects in other activity assays: (a) more efficient inhibition of peptidoglycan synthesis by the penicillin-binding protein (PBP) in vitro, (b) stabilization of the lantibiotic–lipid II complex under TLC conditions, (c) an enhanced ability to immerse into the lipid bilayer and form pores in bacterial and liposomal membranes [52]. However, the clearly noticeable enhancing effect of the Ca^2+^ addition in the PBP inhibition test with the α-component of lacticin 3147 disappeared after complementing it with an equimolar quantity of the β-component [52]. We observed a similar picture in the antimicrobial activity assays of lichenicidin and its α-component: a pronounced effect of Ca^2+^ when testing the individual Lchα and no effect for the equimolar mixture Lchα:Lchβ. It seems plausible that in some situations, Ca^2+^ compensates for the absence of the β-component by mimicking its action. This effect appeared with only one of two tested strains, indicating that Ca^2+^ is an optional cofactor for the lichenicidin antibacterial activity.

The question of which structural and functional features of mersacidin-like peptides and their molecular target are responsible for Ca^2+^ dependence of the mechanism of action remains open. It has been speculated that Ca^2+^ ions are needed to form a salt bridge between the deprotonated Glu17 residue of the lipid II binding motif in mersacidin and the negatively charged groups of lipid II (pyrophosphate, side chain of γ-D-Glu2, or *C*-terminus of D-Ala5), unless it directly interacts with the positively charged ε-amino group of Lys-3 of the pentapeptide [53]. It has been also suggested that the Ca^2+^ effect is the most pronounced for lantibiotics with a neutral net charge [52]. It can be assumed that the Ca^2+^ dependence is due to the specific distribution of charged groups near the lantibiotic binding site in the lipid II molecules. For example, the third position in the pentapeptide of *M. luteus* (like in most Gram-positive bacteria) is occupied by Lys residue [15], while the pentapeptide of *Bacilli* (and also *L. monocytogenes*, *Mycobacteria*, and most Gram-negative species) contains meso-diaminopimelic acid (mDAP) [54,55], which has a carboxyl group attached to C_ε_. The presence of an additional negative charge may necessitate the participation of Ca^2+^ as a bridging ion to form a bond between the two molecules. Electrostatic interactions can also be affected by amidation of mDAP, D-Glu2, and acidic residues that occur in some interpeptide cross-bridges.

Our results deepen the understanding of the mechanisms of action of lipid II-targeting lantibiotics, while at the same time indicating that some aspects of this process remain unclear and require further study.

## 4. Materials and Methods

### 4.1. Lipid II Preparation

Lipid II was synthesized in vitro employing membrane preparations of *Micrococcus luteus* DSM 1970 and cell extracts from *Staphylococcus simulans* 22 containing the soluble cell wall precursor UDP-*N*-acetylmuramic acid pentapeptide (UDP-MurNAc-PP).

UDP-MurNAc-PP was prepared from *S. simulans* 22 as described by Kohlrausch and Höltje [56] and modified by Brötz et al. [57]. In short, *S. simulans* 22 was grown in Mueller–Hinton broth to an OD_600_ of 0.75. Subsequently, 130 µg/mL of chloramphenicol were added. After 15 min, 5 µg of vancomycin were added to each mL of culture, and the cells were harvested after 60 min of further incubation. The resulting cell pellet was resuspended in 20 mL of MilliQ water. UDP-MurNAc-PP was extracted from the cell suspension with two volumes of boiling MilliQ water under stirring. The supernatant of the extraction was lyophilized and used as crude substrate for the lipid II synthesis.

Membrane preparations of *M. luteus* DSM 1970 were generated as described in Breukink et al. [58] and Schneider et al. [59]. Bacterial membranes were obtained by treatment with lysozyme and subsequent centrifugation and stored at −70 °C after two washing steps.

For synthesis of mg amounts of lipid II, 1–2 mL of membrane preparation and crude substrate were incubated in presence of 2.5 µmol of undecaprenyl phosphate (C55-P) and 25 µmol of UDP-GlcNAc in 60 mM Tris-HCl buffer including 5 mM MgCl_2_ pH 8, containing 0.5% of Triton X-100 in a total volume of 10 mL. After one hour at 30 °C the bactoprenol-containing products were extracted using *n*-butanol / 6 M pyridine-acetate, pH 4.2. Thereafter, lipid II was purified on a DEAE cellulose column (HiTrap DEAE FF, 5 mL BV, Amersham Biosciences) by high pressure liquid chromatography (HPLC) and analyzed for the presence of lipid II by thin layer chromatography and MALDI-TOF mass-spectrometry (Appendix A) in comparison to a lipid II reference sample. The lipid II concentration of pooled lipid II-containing samples was determined by the phosphate content according to Rouser [60]. For future analysis, purified lipid II samples were stored after lyophilization.

### 4.2. Antimicrobial Peptide Preparations

The recombinant beta-hairpin peptide AA139 was expressed in *E. coli* BL21 (DE3) under control of T7 promoter as a His8-tagged thioredoxin-containing fusion protein and purified using immobilized metal affinity chromatography, CNBr cleavage, and reversed-phase HPLC [61]. The unlabeled Lchα and Lchβ were isolated from *Bacillus licheniformis* B-511 strain provided by VKM (All-Russian Collection of Microorganisms) using previously published protocols [18].

To prepare the ^15^N-labeled Lchα, the culture of *B. licheniformis* B-511 was grown on LB agar at 37 °C for 16 h, then transferred to 20 mL of M9 minimal medium (4.8 mM Na_2_HPO_4_, 2.2 mM KH_2_PO_4_, 0.85 mM NaCl, 1.87 mM NH_4_Cl) supplemented with 20 mM glucose, 2 mM MgSO_4_, 0.1 mM CaCl_2_, 0.001% thiamin, 10 µM FeCl_3_, 0.000001% tryptone, 0.000001% yeast extract. After 18 h of growth at 37 °C and 220 rpm in an orbital shaker, bacteria were transferred to 20 mL of the same medium with NH_4_Cl substituted by ^15^N-NH_4_Cl (CIL, Tewksbury, MA, USA), and the incubation was repeated. Aliquots of approximately 1 mL of overnight culture were inoculated into 2 L Erlenmeyer flasks, each filled with 160 mL of the same 15N-labeled medium. The flasks were incubated at 37 °C and 220 rpm for 18 h.

At the end of cultivation, the bacterial cells were precipitated by centrifugation at 8000× *g* for 30 min. A total of 400 mL of n-butanol were added to 1 L of the supernatant, and a single extraction step was carried out, after which the aqueous phase was removed using a separating funnel, and the resulting organic extract was evaporated to dryness. The dry extract was redissolved in 50 mL of buffer A1 (30 mM ammonium acetate, pH 5.6, 30% acetonitrile) and applied to a column with Diasorb-100-C8 (2.5 × 10 cm) (BioChemMack-ST, Moscow, Russia) equilibrated with buffer A1 at a flow rate of 2 mL/min. After washing the column with 100 mL of buffer A1, the substances bound to the sorbent were eluted with 50 mL of buffer B1 (30 mM ammonium acetate, pH 5.6, 80% acetonitrile) at a flow rate of 2 mL/min. The resulting eluate was evaporated on a rotary evaporator to dryness and redissolved in 5–6 mL of 50% methanol. The target peptide was purified by reversed-phase HPLC using Xterra Prep RP-18 column (Waters, Milford, MA, USA) equilibrated with solution A2 (5% acetonitrile, 60% methanol) in a linear gradient of solution B2 (35% acetonitrile, 60% methanol) from 0 to 100% for 40 min (2.5% min^−1^) at a flow rate of 0.5 mL/min (Appendix A). Fractions with peak optical absorption at 214 nm were collected and analyzed by MALDI-TOF Reflect III mass spectrometer (Bruker, Karlsruhe, Germany) equipped with a 336 nm UV-laser using 2,5-dihydroxybenzoic acid (Sigma-Aldrich, St. Louis, MO, USA) as a matrix (Appendix A).

### 4.3. Antibacterial Activity Assay

The inhibitory activity of natural Lchα and Lchβ and recombinant AA139 against *Bacillus megaterium* VKM41 and *Micrococcus luteus* B1314 was determined as described previously [18]. Half Mueller–Hinton broth (½ MH) (Sigma-Aldrich, St. Louis, MO, USA), without salts or the same broth with the addition of 1.25 mM MgCl_2_ or 1.25 mM CaCl_2_, was used for serial dilution experiments with the final culture concentration 5 × 10^6^ CFU mL^−1^. The t-test was performed to study the statistical difference (*p*-value) between the means of controls and samples with different peptide concentrations.

### 4.4. NMR Experiments and Data Analysis

0.3 mM samples of the ^15^N-labeled Lchα were used for the NMR study in water, *d3*-methanol, and in *d38*-DPC micelles environment. Perdeuterated *d38*-DPC (CIL, Tewksbury, MA, USA) was added to the Lchα sample in water using concentrated stock solution. The final detergent to peptide molar ratio (D:P) was in the range of 30:1–90:1. The pH of the samples was adjusted to 4.0 (water), 3.5 (methanol, uncorrected pH-meter readings measured using glass body micro pH combination electrode, Sigma-Aldrich, St. Louis, MO, USA), or 5.8 (DPC) using concentrated HCl or NaOH solutions. An amount of 5% D_2_O was added to the water and DPC samples. NMR spectra were measured at 30 °C (water), 27 °C (methanol), or 45 °C (DPC) on AVANCE-III 600 and AVANCE-III 800 spectrometers equipped with cryogenically cooled probes (Bruker, Karlsruhe, Germany).

^1^H and ^15^N resonance assignments were obtained by a standard procedure using a combination of the 3D ^1^H-TOCSY–^15^N-HSQC and ^1^H-NOESY–^15^N-HSQC, and 2D ^1^H-TOCSY and ^1^H-NOESY spectra measured with ^15^N decoupling. Spectra were analyzed in the CARA program (version 1.84, Zurich, Switzerland). ^1^H chemical shifts were measured relative to the residual H_2_O signal at 4.75 ppm (30 °C) or 4.60 ppm (45 °C). Spectra in methanol were referenced relative to residual CH_3_ signal at 3.31 ppm. ^13^C resonance assignments were obtained using 2D ^13^C-HSQC spectra. ^13^C and ^15^N chemical shifts were referenced indirectly.

The spatial structure calculation was performed in the CYANA (version 3.97) program. Structural restraints (interproton distances, torsion angles, and hydrogen and thioether bonds) were taken from a previous publication [18]. Relaxation parameters (R_1_, R_2_, ^15^N-{^1^H}-NOE) of ^15^N nuclei were obtained using the standard set of ^15^N-HSQC-based pseudo 3D experiments measured at 800 MHz.

A 1.8 mM sample of the lipid II was used for the NMR study in DPC micelles environment. The dry lipid II was dissolved in the DPC micelles solution (90 mM, detergent to lipid II molar ratio of ~50:1). The pH of the sample was adjusted to 5.8 using concentrated HCl or NaOH solutions, and 5% D_2_O was added. NMR spectra were measured at 45 °C on AVANCE-III 600 and AVANCE-III 800 spectrometers equipped with cryogenically cooled probes. The lipid II resonance assignment was obtained using a standard approach using a combination of 2D ^1^H-TOCSY (τ_m_ = 80 ms), ^1^H-NOESY (τ_m_ = 150 ms), and ^13^C-HSQC spectra. For study of Lchα interaction with lipid II in the DPC micelles environment, the sample containing 0.3 mM of ^15^N-Lchα (10 mM DPC) was titrated with 1.8 mM lipid II sample (90 mM DPC). The final Lchα:lipid II:DPC molar ratio was of ~1:4:240 (pH 5.8). For study of the Lchα interaction with Ca^2+^, the obtained Lchα/lipid II/DPC sample was titrated with CaCl_2_ solution. The final concentration of Ca^2+^ was 3.5 mM.

The Lchα binding to the DPC micelle was quantified using the chemical shift titration data assuming fast (on the NMR time scale) exchange of the peptide molecules between solution and the micelles [41]. In this case, the observed chemical shift of a nuclei (*δ^obs^*) depends on the chemical shift in the bound state (*δ^b^*), the chemical shift in the free state (*δ^f^*), and on the bound peptide (*C_b_*) to free peptide (*C_f_*) ratio:(1)$$\delta^{obs}=\frac{\delta^{f}\times C_{f}+\delta^{b}\times C_{b}}{C_{0}}$$
where *C*_0_ is the total peptide concentration.

This isotherm was analyzed using Langmuir Equation (2) with two independent parameters: free energy of binding (Δ*G*^0^) and number of DPC molecules (*N*) that form the site of the peptide binding. The parameter *K_N_* represents the affinity constant of the peptide to the site on the surface of DPC micelle.
(2)$$Exp\left(\frac{\Delta G^{0}}{RT}\right)=\frac{1}{K_{N}}=\frac{C_{f}\left(\left[DPC\right]-N\times C_{b}\right)}{N\times C_{b}}$$

The equilibrium dissociation constant of the Lchα/lipid II complex (*K_D_*) was determined assuming 1:1 stoichiometry of the complex. The dependence of ^1^H^15^N Asn32 signal intensity on the lipid II concentration was analyzed assuming a slow (on the NMR time scale) exchange of the peptide between the free and lipid II bound states. In this case, the concentrations of the bound and free peptide (*C_b_* and *C_f_*, respectively) are proportional to the intensities of corresponding cross-peaks in the ^15^N-HSQC spectrum. The proportionality factor depends on the relaxation rates and is different for different signals. The binding reaction goes in the lipid phase, so all concentrations were recalculated relative to DPC.
(3)$$K_{D}=\frac{C_{f}}{C_{b}}\times \frac{{\left[Lipid\;II\right]}_{0}-C_{b}}{\left[DPC\right]}$$
where [*lipid II*]_0_ is the total lipid II concentration in the sample.

The nonlinear fitting of Equations (2) and (3) to the experimental data was performed with Mathematica software (Wolfram Research, Champaign, IL, USA). The dilution of the samples during titrations was taken into account.

### 4.5. Molecular Modeling

MD simulations were performed using the GROMACS package, version 2020.4 [62], and modified Gromos 43a2x force field [63]. Simulations were carried out with 2 fs integration time step and imposed 3D periodic boundary conditions. SPC water model and an improved version of lipid parameters introduced by Berger et al. were used [64]. A 12 Å spherical cutoff function was used to truncate van der Waals interactions. Electrostatic effects were treated using the particle-mesh Ewald summation (real space cutoff 12 Å) [65]. The constant temperature (315 K) and pressure (1 bar) of the systems were maintained by the V-rescale thermostat [66] and Berendsen coupling method [67]. A compressibility of 4.5 × 10^−5^ bar^−1^ for the barostat was applied. A certain number of solvent molecules were replaced by Na^+^ or Cl^−^ ions for restoring electroneutrality of each system. Prior to calculating the MD trajectory, all systems were subjected to energy minimization (1000 conjugate gradients steps), and then the temperature in the systems was increased linearly from 5 to 315 K during 300-ps MD run with fixed heavy atoms of the peptide, dimethyl pyrophosphate (DMPPi), and/or lipid II. Studied molecules, bilayer lipids and water molecules were coupled separately.

In this study, we performed a series of MD simulations for the following systems:**Full-length Lchα in the presence of DMPPi**, which mimics the lipid II pyrophosphate group, in water solution. The initial conformation of Lchα was retrieved from the recalculated NMR structure with corrected chirality of Cβ atom of Abu residues. The force-field parameters for the non-canonical amino acids such as unsaturated Dha, Dhb, lanthionine, and methyllanthionine were taken from our previous work on nisin [21]. The *N*-terminal 2-oxobutyryl group was replaced with the standard aliphatic valine residue. The peptide was solvated with water, and one or three DMPPi ions were randomly placed in a cubic box with a minimum distance of 7 Å to Lchα, in order to determine the preferable Lchα binding site, as well as the mutual adaptation of the peptide and its target.**Truncated Lchα analogs containing potential lipid II binding sites:** nisin-like *N*-terminal Lchα_1–21_ fragment and mersacidin-like *C*-terminal Lchα_22–32_ fragment in the presence of DMPPi in water. The starting coordinates for Lchα_1–21_ and Lchα_22–32_ for MD calculations were extracted from equilibrated states of the full-length molecule. Several Ca^2+^-containing trajectories were calculated for this and the previous group of simulations to investigate the role of this ion in intermolecular interactions of Lchα (see Table 1).**Full-length Lchα in the presence of the hydrated lipid bilayer** (with the peptide initially randomly placed outside the bilayer interfacial region): each bilayer leaflet contained 144 lipid molecules and was composed of the palmitoyl-oleoyl-phosphatidylglycerol(POPG)/palmitoyl-oleoyl-phosphatidylethanolamine (POPE) mixture at a ratio of 3:1, which mimics the Gram-positive bacteria membrane [68]. The semi-isotropic pressure coupling in the bilayer plane and along the membrane normal was used in the simulation. MD-averaged values of the area per lipid and the POPG/POPE bilayer thickness were 54.8  ±  0.3 Å^2^ and 39.2  ±  0.4 Å, respectively.

**Complex of full-length Lchα with lipid II embedded into the hydrated model bacterial membrane:** the complex of Lchα with lipid II in the membrane was assembled according to the previously published procedure [21]. The representative *N*-terminal and *C*-terminal complex structures from the Lchα/DMPPi solvent simulations were projected to the bilayer-bound lipid II via its superimposition with suitable lipid II conformations (root-mean-square deviation (RMSD) between coordinates of the pyrophosphate group in lipid II and DMPPi is less than 0.2 Å). The 1000-ns MD trajectories of membrane-bound lipid II were taken from our previous work [21]. Among possible models of the Lchα/lipid II complex in the membrane, doubled in view of DMPPi symmetry, only five conformers for the *N*-terminal binding domain were taken for further investigation. In these structures, lipid head groups do not intersect the peptide. A slightly modified protocol was used for reconstruction of the complex of the *C*-terminal Lchα domain with lipid II in the membrane. In this case, the restriction on the intersection of peptide atoms with lipid head groups has been weakened in favor of the larger number of intermolecular bonds. This protocol yielded a set of complexes stable in the MD simulations.

Unrestrained MD simulation of each system was repeated 3–5 times by random assignment of initial velocities. All systems studied are listed in Table 1.

Analysis of MD trajectories was performed using utilities from the GROMACS package (root-mean-square deviation (RMSD) from the starting structure, inter-, and intramolecular hydrogen bonds profiles and dihedral angles) and the custom IMPULSE program [69]. Preferred conformations of lichenicidin’s regions were found using the GROMACS *cluster* module, where backbone atoms of residues 3–21 for the *N*-terminus and 22–32 for the *C*-terminus were superimposed. A cut-off of 1.5 Å was applied, and the largest clusters (>10%) were extracted.

### 4.6. Accessing Codes

Experimental restraints and atomic coordinates for the Lchα structure in methanol solution have been deposited in PDB under accession code 8C5J.

## Figures and Tables

**Figure 2 ijms-24-01332-f002:**
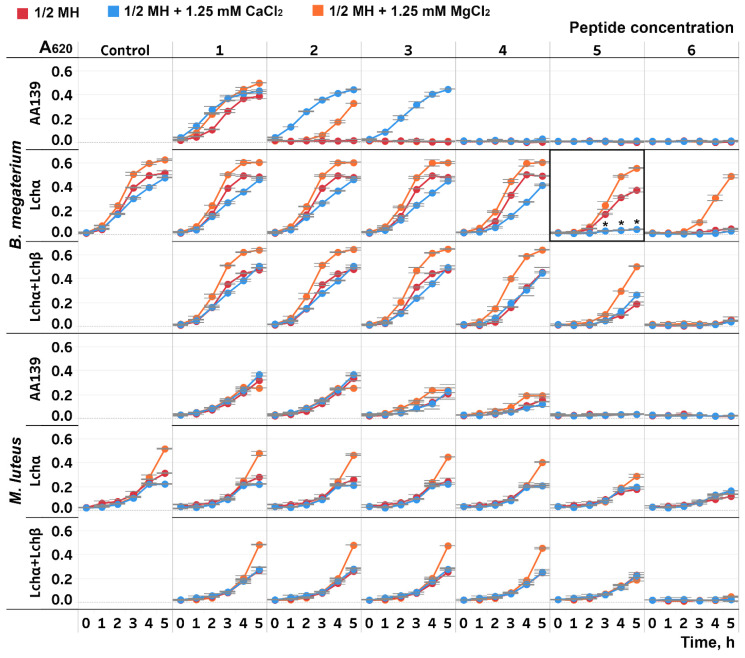
Growth curves of the Gram-positive bacteria *Bacillus megaterium* VKM41 and *Micrococcus luteus* B1314 strains in the presence of Lchα, Lchα/Lchβ mixture (1:1), and the β-hairpin peptide AA139. The concentrations used (1–6): Lchα—2.5, 5, 10, 20, 30, and 40 μg/mL; Lchα + Lchβ—0.19, 0.38, 0.75, 1.5, 3, and 6 μg/mL (total concentration); AA139—0.16, 0.31, 0.63, 1.25, 2.5, and 5 μg/mL. Control—the growth curves of the cultures without the peptides. *Red*, *orange*, and *blue* lines correspond to the growth curves of the cultures in half Mueller–Hinton (½ MH) medium without salts and with the addition of 1.25 mM MgCl_2_ or 1.25 mM CaCl_2_, respectively. Error bars represent standard deviation between technical replicates. The most pronounced increase in the Lchα antimicrobial activity in the presence of Ca^2+^ was observed at the peptide concentration of 30 μg/mL ≅ 9 μM (enclosed in frame; differences in mean A values between the wells without salts and with the addition of 1.25 mM CaCl_2_ were compared by *t*-test (significance level is * *p* < 0.05)).

**Figure 3 ijms-24-01332-f003:**
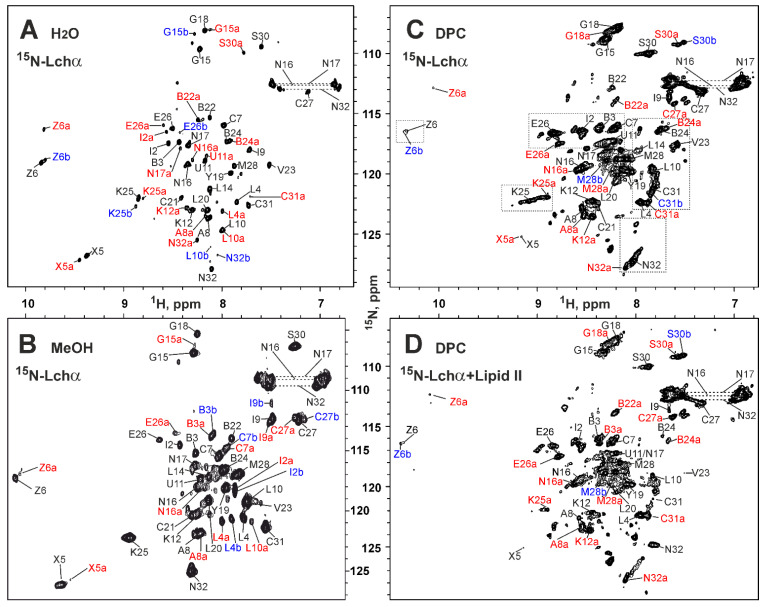
2D ^15^N-HSQC spectra and resonance assignment of Lchα in different environments. (**A**) H_2_O (Lchα 0.3 mM, pH 4.0, 30 °C). (**B**) *d3*-methanol (Lchα 0.3 mM, pH 3.5, 27 °C). (**C**) DPC micelles solution (Lchα 0.35 mM, DPC 30 mM, D:P = 86:1, pH 5.8, 45 °C). (**D**) DPC micelles solution with lipid II (Lchα 0.18 mM, lipid II 0.72 mM, DPC 42 mM, Lchα:lipid II:DPC = 1:4:240, pH 5.8, 45 °C). The resonance assignment is shown. The signals of the major and two minor forms of the peptide are marked in black, red, and blue, respectively. The signals of the minor forms are additionally marked with “a” or “b”. The residue names are given in the one letter code format, where 2,3-didehydroalanine, 2,3-didehydrobutyrine, lanthionine, and methyllanthionine are abbreviated as X, Z, U-C, and B-C, respectively. Areas highlighted by rectangles in panel (**C**) are discussed in Section 2.5.

**Figure 4 ijms-24-01332-f004:**
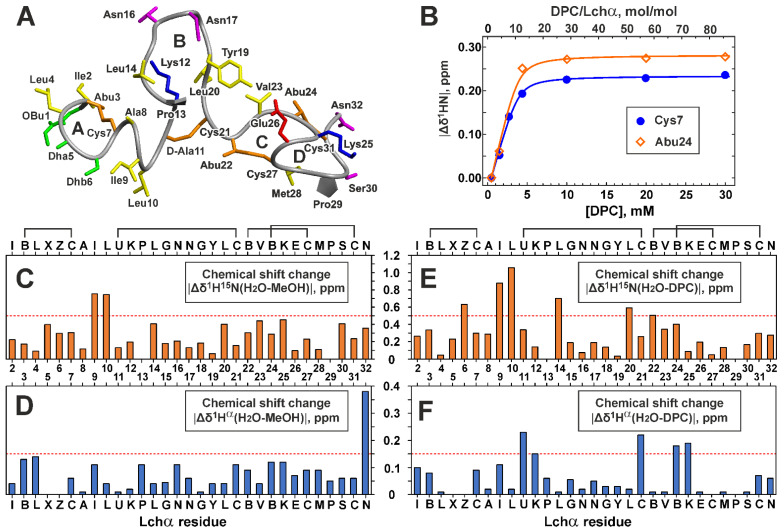
Spatial structure of Lchα in methanol (**A**), Lchα binding to DPC micelles (**B**), and comparison of Lchα chemical shifts in different environments (**C**–**F**). (**A**) The representative conformer of Lchα in ribbon representation. The positively charged, negatively charged, hydrophobic/aromatic, polar, and non-standard residues are colored in *blue*, *red*, *yellow*, *magenta*, and *green*, respectively. The lanthionine and methyllanthionine bridges are colored *orange*. Thioether bridging rings are marked with the capital letters A, B, C, and D. (**B**) Titration of 0.35 mM Lchα sample by DPC. Titration curves for ^1^HN of Cys7 and Abu27 are approximated by Langmuir isotherm. (**C**–**F**) Changes in ^1^H^15^N (Δδ=(ΔδH1)2+(ΔδN15/5)2) and ^1^H^α^ chemical shifts upon Lchα transfer from water to methanol (**C**,**D**) and DPC micelles (**E**,**F**). The residue names are given in the one letter code format (see caption to Figure 3).

**Figure 5 ijms-24-01332-f005:**
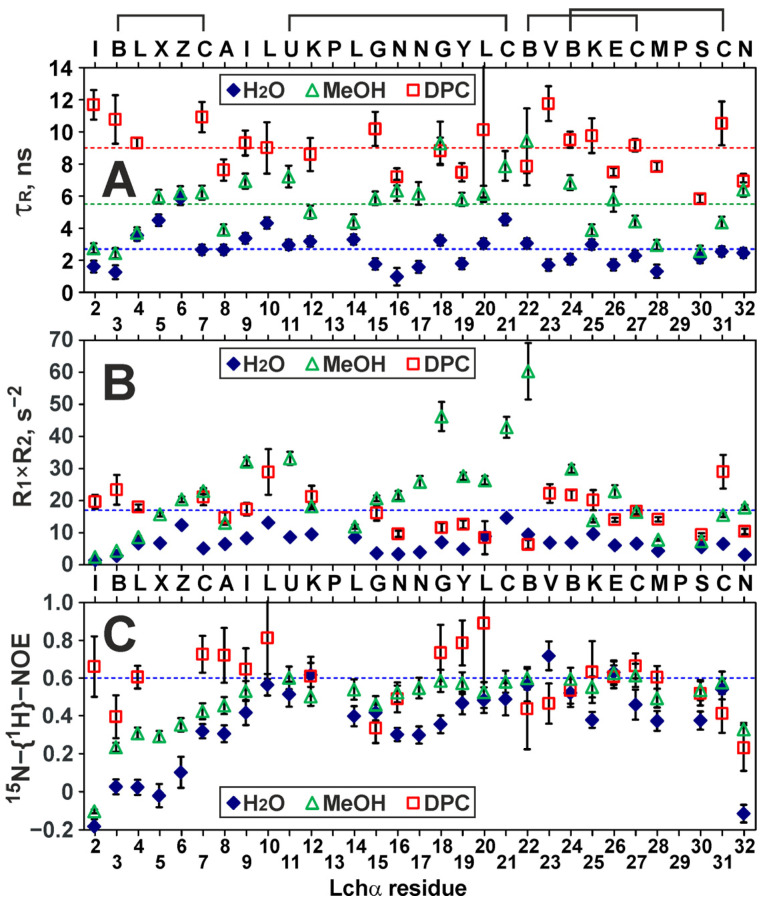
^15^N relaxation measurements at 80 MHz revealed the dynamic properties of Lchα in water, methanol, and DPC micelles. (**A**) The apparent rotational correlation times were calculated from the R_2_/R_1_ ratio. The levels corresponding to the average values (2.7, 5.5, 9.0 ns, respectively) are shown by *broken lines*. (**B**) Calculated R_1_×R_2_ products. The residues displaying R_1_ × R_2_ > 17 s^−2^ are subjected to exchange fluctuations on the μs–ms time scale [42]. (**C**) Steady-state ^15^N-{^1^H}-NOE. The residues displaying NOE < 0.6 are subjected to extensive motions on the ps–ns time scale. The data in the figure represent the results obtained from one series of NMR experiments. Each NMR experiment includes multiple signal averaging. Error bars correspond to experimental errors estimated from the noise level in the spectra.

**Figure 6 ijms-24-01332-f006:**
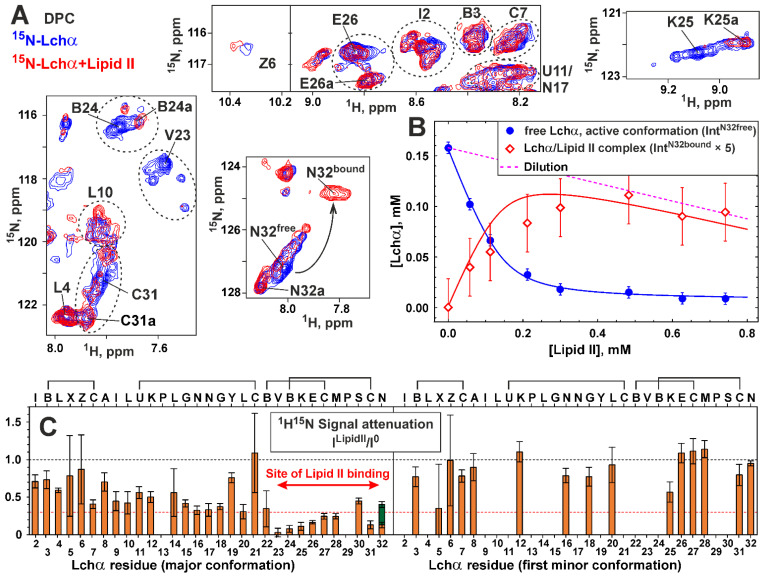
NMR data defines the lipid II binding to Lchα in the DPC micelles environment. (**A**) Fragments of 2D ^15^N-HSQC spectra of Lchα (0.3 mM, pH 5.8, 45 °C) measured in the absence (*blue contours*) and the presence of 4:1 molar excess of lipid II (*red contours*). The signals of the first minor form are labeled with “a”. The residue names are given in the one letter code format (see caption to Figure 3). Corresponding fragments of the ^15^N-HSQC spectrum are highlighted by rectangles in Figure 3C. (**B**) Curves describing lipid II binding to Lchα. *Blue circles*/*curve*—concentration of the active conformation of Lchα non-bound to lipid II determined from the intensity of Asn32^free 1^H^15^N signal and approximated by the binding model (Equation (3)). *Red curve*—concentration of the Lchα/lipid II complex determined from the fitted binding model. *Red diamonds*—concentration of the Lchα/lipid II complex determined from the intensity of Asn32^bound 1^H^15^N signal. Intensity of Asn32^bound^ signal was increased fivefold to account for the difference in relaxation. (**C**) Attenuation of ^1^H-^15^N-HSQC signals of the major and minor Lchα forms induced by addition of 4:1 molar excess of lipid II. The 0.3 *threshold line* subdivides data points in two groups: the residues interacting with lipid II (*below*) and not (*above*). Two bars are shown for Asn32 residue corresponding to Asn32^free^ (*orange*) and Asn32^bound^ (*dark green*) signals. Sample dilution is taken into account. The data in the figure represent the results obtained from one series of NMR experiments. Each NMR experiment includes multiple signal averaging. Error bars correspond to experimental errors estimated from the noise level in the spectra.

**Figure 7 ijms-24-01332-f007:**
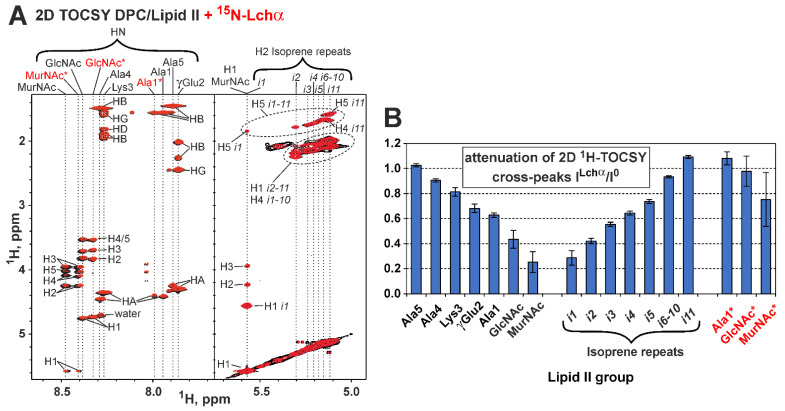
NMR data defines the Lchα binding to lipid II in the DPC micelles environment. (**A**) Fragments of 2D ^1^H-TOCSY spectra of lipid II (pH 5.8, 45 °C) measured in the absence (*black contours*) and the presence of Lchα (*red contours*, lipid II/Lchα = 4:1). Isoprene repeats are labeled as in Figure 1F (*i1* is connected to pyrophosphate). The signals of the impurity (GlcNAc-MurNAc-pentapeptide) are labeled with “*”. (**B**) Attenuation of ^1^H-TOCSY signals of the lipid II and impurity induced by addition of Lchα. The strongest TOCSY cross-peak (in the vertical direction) for each HN or H2 resonances (horizontal direction) was taken for analysis. The data represent the results obtained from one series of NMR experiments. Each NMR experiment includes multiple signal averaging. Error bars correspond to experimental errors estimated from the noise level in the spectra.

**Figure 8 ijms-24-01332-f008:**
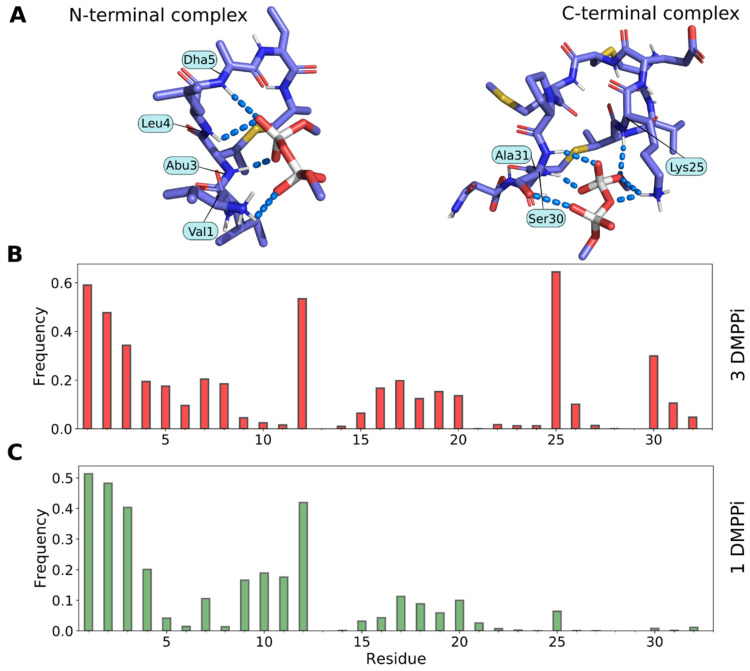
Molecular modeling reveals both the *N*- and the *C*-terminal pyrophosphate recognition sites in Lchα. (**A**) Representative *N*-terminal (*left*) and *C*-terminal (*right*) complexes of Lchα with DMPPi, which formed spontaneously during MD simulations. Interacting residues of the peptide are *subscribed*. (**B**,**C**) Per-residue Lchα/DMPPi hydrogen bonding patterns observed in MD trajectories containing three (**B**) or one (**C**) DMPPi ion. Bar height is the lifetime of the hydrogen bond between a given residue and DMPPi as a fraction of the MD time (estimated based on five 500-ns trajectories). Note that the *C*-terminal site is active only when the *N*-terminal site is already occupied (upon DMPPi excess). Data are given for the full-length Lchα; similar results were obtained for the *N*-terminal Lchα_1–21_ and *C*-terminal Lchα_22–32_ fragments (data not shown). Please note that the *N*-terminal 2-oxobutyryl group was replaced with the valine residue.

**Figure 9 ijms-24-01332-f009:**
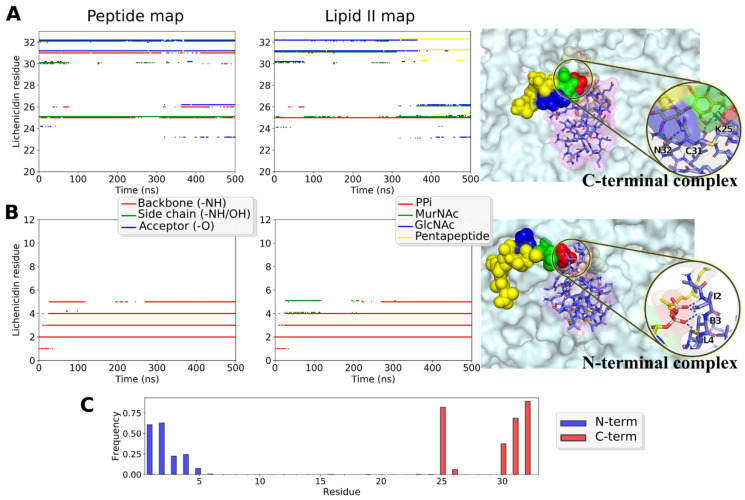
Structure and dynamics of the predicted *C*- and *N*-terminal complexes of the full-length Lchα with lipid II in the model bacterial membrane. (**A**,**B**) Intermolecular interactions and the snapshot of the complexes’ structure from MD simulations. *Left* and *Middle panels*: intermolecular hydrogen bonding maps that describe complex organization and dynamics. *Left panel:* peptide groups taking part in h-bond formation at a certain MD time: backbone amide group that donates the proton (*red dots*), side chains (*green*), and backbone carbonyl group, which accepts the proton (*blue*). *Middle panel:* the lipid II groups forming the same h-bonds: PPi (*red*), MurNAc sugar (*green*), GlcNAc sugar (*blue*), and the pentapeptide (*yellow*). *Right panel:* the representative snapshot from MD. Lipid II is color-coded as in the *middle panel*. The membrane is shown with the *surface*. Interactions are zoomed in the *inset*. (**A**) *C*-terminal complex involves the peptide side chain. Note that we obtained several unlike modes of the *C*-terminal complex, relatively stable in MD (Appendix A). (**B**) The *N*-terminal complex, in a nisin-like manner, is based exclusively on the backbone amide protons interactions with the pyrophosphate moiety. (**C**) Lchα residues involvement into the *N*- (*blue*) and *C*-terminal (*red*) complexes, as shown with h-bonds lifetime (as a fraction of cumulative time of MD sets).

**Table 1 ijms-24-01332-t001:** Molecular dynamics simulations conducted in this work.

System Composition	MD Length (ns)	Number of Trajectories
*Lichenicidin with DMPPi in solution*		
Lchα/DMPPi_1_/Water_10336_/Na^+^_1_	500	5
Lchα/DMPPi_3_/Water_9704_/Na^+^_5_	500	2
Lchα/DMPPi_3_/Water_10321_/Ca^2+^_3_/Cl^−^_1_	500	3
Lchα_1–21_/DMPPi_3_/Water_7648_/Na^+^_5_	500	3
Lchα_1–21_/DMPPi_3_/Water_7648_ /Ca^2+^_3_/Cl^−^_1_	500	2
Lchα_22–32_/DMPPi_3_/Water_9242_/Na^+^_6_	500	2
Lchα_22–32_/DMPPi_3_/Water_10216_/Ca^2+^_3_	500	3
*Lichenicidin in the membrane environment*		
Lchα/POPG_126_/POPE_96_/Water_20358_/Na^+^_191_	500	3
*Lichenicidin/lipid II complex in bacterial membrane*		
*C-terminal complex*		
Lchα/Lipid II/POPG_186_/POPE_66_/Water_11966_/Na^+^_188_	500	3
*N-terminal complex*		
Lchα/Lipid II/POPG_186_/POPE_66_/Water_11985_/Na^+^_188_	500	5

## Data Availability

All data presented in this study are available from the corresponding author on reasonable request.

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
