# Peer review of "Specific Binding of the α-Component of the Lantibiotic Lichenicidin to the Peptidoglycan Precursor Lipid II Predetermines Its Antimicrobial Activity"

_ijms, 2023, doi:10.3390/ijms24021332_

Round 1

Reviewer 1 Report

This paper is written according the rules. Everything is clear described and explained. The only thing is that the section of Introduction is too long.

Author Response

Dear Reviewer,
Thank you very much for your feedback!
Point 1: The only thing is that the section of Introduction is too long.
Response 1: We've shortened the introduction by a couple of sentences. The rest, in our opinion, is important for understanding the background of our work.

Reviewer 2 Report

I have read the manuscript entitled "Specific binding of the α-component of the lantibiotic lichenicidin to the peptidoglycan precursor lipid II predetermines its antimicrobial activity" with great interest and I think it is in principle suited for a publication in IJMS. It is well-written, well-constructed and clearly exposed. Authors have used theoretical and experimental methods to demonstrate the understanding of the mechanisms of action of lipid II-targeting lantibiotics.

However, I also have
minor comments and some questions:

Lines 96-99. I think that Glu26 is not discussed in the cited articles [22-24]. Please check.

Lines 155-157. Also the peptide AA139 is not investigated in the cited work [38]. If necessary, replace with a more suitable reference (e.g. DOI: 10.1016/j.bbagen.2022.130156.) 

Lines 354-355. “The characteristic time of this process is >> 6 ms (estimated from frequency difference of Asn32 signals)”. Please provide the information to show how this assessment was made.

Lines 571-573. Please check the text and relevance of the reference [20], as that work considers effects only for Asn20 and Met21 residues, but not for the Asn20-Met21-Lys22 tripeptide.

Lines 649-651. “It was also suggested that the Ca2+ effect is the most pronounced for lantibiotics with a neutral or negative net charge [49]”. It seems that in [49] it was suggested only for “neutral net charge”. Please check.

Line 696. Please explain: what is an “IMAC”? Immobilized metal affinity chromatography?

Lines 702-703. “ 0.000001% tryptone, 0.000001% yeast extract”. Please check the concentrations.

Line 738. “… uncorrected pH-meter readings”. Please check.

Figures 5, 6, and 7. Do the error bars represent standard deviation between technical replicates?

Author Response

Dear Reviewer,
Thank you for the time and effort that you have dedicated to carefully reading our work. We appreciate your valuable feedback and have revised the manuscript to reflect the suggestions you provided.

Point 1: Lines 96-99. I think that Glu26 is not discussed in the cited articles [22-24]. Please check.
Response 1: Corrected.

Point 2: Lines 155-157. Also the peptide AA139 is not investigated in the cited work [38]. If necessary, replace with a more suitable reference (e.g. DOI: 10.1016/j.bbagen.2022.130156.)
Response 2: The reference to the original work on AA139, which disappeared under mystical circumstances, has been restored: doi:10.1038/s41467-020-16950-x

Point 3: Lines 354-355. “The characteristic time of this process is >> 6 ms (estimated from frequency difference of Asn32 signals)”. Please provide the information to show how this assessment was made.
Response 3: The detailed description was added to the text.

Point 4: Lines 571-573. Please check the text and relevance of the reference [20], as that work considers effects only for Asn20 and Met21 residues, but not for the Asn20-Met21-Lys22 tripeptide.
Response 4: Added references to papers showing the effect of Lys22 mutations in the hinge region on antimicrobial activity: 10.1007/s00253-004-1599-1, 10.1111/j.1365-2958.2008.06279.x.

Point 5: Lines 649-651. “It was also suggested that the Ca2+ effect is the most pronounced for lantibiotics with a neutral or negative net charge [49]”. It seems that in [49] it was suggested only for “neutral net charge”. Please check.
Response 5: Changed to "lantibiotics with a neutral net charge".

Point 6: Line 696. Please explain: what is an “IMAC”? Immobilized metal affinity chromatography?
Response 6: Substituted "IMAC" with "Immobilized metal affinity chromatography"

Point 7: Lines 702-703. “ 0.000001% tryptone, 0.000001% yeast extract”. Please check the concentrations.
Response 7: These concentrations are correct. We add "homeopathic" quantities of rich medium components (along with some metals and thiamin) to enhance the bacterial growth in 15N-labeled M9 medium. The amount of the introduced 14N isotope should be very small so as not to affect the quality of the resulting labeled peptide.

Point 8: Line 738. “… uncorrected pH-meter readings”. Please check.
Response 8: The pH value is defined through autoprotolysis constant. For water this constant (KW) is given by KW = [H+][OH-] = 10^-14 (at 25 °C). In this case the neutral pH is defined as the state at which [H+] = [OH-], which occurs when [H+]=10^-7 or at pH = 7.
At the same time, for methanol the autoprotolysis constant is Km = [H+][CH3O-] = 10^-16.6 and neutral state is at pH = 8.3. Therefore, the pH-meter readings in methanol does not give the correct pH of the sample. To compare conditions used in different works performed in our and other labs, we measure the pH of the samples in methanol by conventional electrode and pH-meter as if we had a normal water solution. In our publications, we report the obtained pH values as uncorrected pH-meter readings. So, conditions of our experiments can be easily reproduced in other labs.
To give more details of our experiments we provide the model of electrode used for pH measurements.

Point 9: Figures 5, 6, and 7. Do the error bars represent standard deviation between technical replicates?
Response 9: These figures describe the results of one (or one series of) NMR experiment(s). Each NMR experiment includes multiple signal averaging. The error bars correspond to experimental errors estimated from the noise level in the spectra. This information was added to the figure captions.

Reviewer 3 Report

This article is very good. The data is very sufficient. I have found only a few minor problems.

1. Too many spaces between the last two words in Abstract .

2. The name of the bacteria in the keyword should be changed to italicized.

3. Line 106, "2" in "H2O" shall be the subscript.

4. Line 170, "2 +" in "Ca2 +" shall be superscript.

5. Line 811-829, "1"  "2"  "3" in subtitle is expressed in other ways, such as "a"  "b"  "c".

Author Response

Dear Reviewer,
We appreciate your feedback and thank you for the time you have dedicated to carefully reading our work. We have revised the manuscript and incorporated the suggested change into the text.

Point 1: Too many spaces between the last two words in Abstract.
Response 1: Corrected.

Point 2: The name of the bacteria in the keyword should be changed to italicized.
Response 2: Corrected.

Point 3: Line 106, "2" in "H2O" shall be the subscript.
Response 3: Corrected.

Point 4: Line 170, "2 +" in "Ca2 +" shall be superscript.
Response 4: Corrected.

Point 5: Line 811-829, "1"  "2"  "3" in subtitle is expressed in other ways, such as "a"  "b"  "c".
Response 5: For this numbered list, we used the MDPI_3.7_itemize style from the IJMS template for MS Word. It uses numbers instead of letters to designate list items. We believe we should not change the default style settings.